# Rad6 Regulates Conidiation by Affecting the Biotin Metabolism in *Beauveria bassiana*

**DOI:** 10.3390/jof10090613

**Published:** 2024-08-28

**Authors:** Yuhan Guo, Haomin He, Yi Guan, Longbin Zhang

**Affiliations:** Fujian Key Laboratory of Marine Enzyme Engineering, Fuzhou University, Fuzhou 350108, China; 220827012@fzu.edu.cn (Y.G.); hehaomin1027@163.com (H.H.)

**Keywords:** entomopathogenic fungi, *Beauveria bassiana*, Rad6, conidiation, biotin

## Abstract

Rad6 is a canonical ubiquitin-conjugating enzyme known for its role in regulating chromosome-related cellular processes in yeast and has been proven to have multiple functions in *Beauveria bassiana*, including insect-pathogenic lifestyle, UV damage repair, and conidiation. However, previous studies have only reported the key role of Rad6 in regulating conidial production in a nutrient-rich medium, without any deep mechanism analyses. In this study, we found that the disruption of Rad6 leads to a profound reduction in conidial production, irrespective of whether the fungus is cultivated in nutrient-rich or nutrient-poor environments. The absence of *rad6* exerts a suppressive effect on the transcription of essential genes in the central developmental pathway, namely, *brlA*, *abaA*, and *wetA*, resulting in a direct downregulation of conidiation capacity. Additionally, mutant strains exhibited a more pronounced decline in both conidial generation and hyphal development when cultured in nutrient-rich conditions. This observation correlates with the downregulation of the central developmental pathway (CDP) downstream gene *vosA* and the upregulation of *flaA* in nutrient-rich cultures. Moreover, single-transcriptomics analyses indicated that irregularities in biotin metabolism, DNA repair, and tryptophan metabolism are the underlying factors contributing to the reduced conidial production. Comprehensive dual transcriptomics analyses pinpointed abnormal biotin metabolism as the primary cause of conidial production decline. Subsequently, we successfully restored conidial production in the Rad6 mutant strain through the supplementation of biotin, further confirming the transcriptomic evidence. Altogether, our findings underscore the pivotal role of Rad6 in influencing biotin metabolism, subsequently impacting the expression of CDP genes and ultimately shaping the asexual life cycle of *B. bassiana*.

## 1. Introduction

The ubiquitin post-modification system comprises ubiquitin-activating (E1), ubiquitin-conjugating (E2), and ubiquitin-ligating (E3) enzymes, which transfer ubiquitin molecules to substrates [1]. Ubiquitin-marked substrates typically undergo selective degradation, influencing their nature, function, and fate [1,2]. E2 proteins act as a bridge between E1 and E3 enzymes, regulating ubiquitin chain initiation, formation, and elongation. They determine the direction of ubiquitination, ranging from monoubiquitination to multiubiquitination, thereby dictating the fate of the modified proteins [3].

Biotin, also known as vitamin B7, vitamin H, and coenzyme R, is a water-soluble vitamin that interacts directly with a wide range of enzymes and is one of the essential cofactors involved in the central pathways of metabolism in prokaryotic and eukaryotic cells [4]. As one of the vital vitamins for all living organisms, biotin plays a crucial role in many important biological processes, such as biotin-dependent carboxylases, fatty acid biosynthesis, gluconeogenesis and amino acid metabolism, cell signaling, gene expression, and chromatin structure [4,5,6]. The increase or decrease in biotin affects the life activities of organisms, e.g., biotin is vital for the growth, virulence, and spore production of *Alternaria alternata* [7]. Biotin can enhance the osmotic tolerance of *Torulopsis mogii* by promoting the accumulation of alginate, increasing the K^+^/Na^+^ ratio, promoting ATPase activity, and increasing the synthesis of long-chain highly saturated fatty acids [8]. Biotin deficiency affects the expression of genes that synthesize biotin in yeast as well as precursor transport genes and the transcription factor Vhr1p gene [9]. Biotin influences *Candida* proliferative capacity and virulence by regulating the expression of the biotin transcription factor Vhr1 and the biotin transporter protein Vht1 and, thereby, affecting the proliferative capacity and virulence of *Candida* [10].

In filamentous fungi, the central developmental pathway (CDP) plays a pivotal role in influencing fungal asexual development by regulating the expression of downstream conidiation-specific genes through cascade regulation. BrlA, AbaA, and WetA are three key regulators in the CDP [11,12]. In *B. bassiana*, the disruption of *brlA* and *abaA* not only results in the elimination of conidia and blastospore production, but also inhibits its dimorphic transition [13]. WetA and its downstream velvet protein, VosA, are associated with conidial maturation [14]. In addition, *flb* genes, including *flbA-E*, are essential for CDP activation by mediating the transcription of *brlA* and *abaA* [15]. These findings indicate that the CDP has a complex regulatory network and is indispensable in the asexual development of *B. bassiana*.

The *rad6* gene encodes a ubiquitin-conjugating enzyme in the model fungus *Saccharomyces cerevisiae* [16]. The disruption of Rad6 leads to multiple deficiencies in *S. cerevisiae*, affecting DNA repair, selective proteolysis, checkpoints, meiosis, and sporulation [17,18,19]. Entomopathogenic fungi are employed globally as biocontrol agents, yet few researchers have explored the role of the ubiquitin-conjugating enzyme Rad6 in their life cycles. Due to the role of Rad6 in yeast and the multiple functions of biotin in fungi, this raises the following question: would biotin have any relationship with Rad6 in regulating biological potential in *B. bassiana*? We hypothesized that the orthologous Rad6 in *B. bassiana* may play a role with biotin in conidiation similar to that in yeast. Transcriptomics is a widely utilized method in the study of filamentous fungi. Hou et al. applied transcriptomic investigation to reveal the survival mechanism for *B. bassiana* under linoleic acid stress [20]. Wang et al. uncovered the thousands of differentially expressed genes involved in a central developmental pathway and secondary metabolism under light or dark incubation conditions by transcriptomic analysis [21]. This study aimed to test this hypothesis by comparing the phenotypes and transcriptomic patterns of Rad6 mutants and wild types (WTs). As shown below, Rad6 mediates the conidiation capability of *B. bassiana*, possibly through the regulation of biotin metabolism.

## 2. Materials and Methods

### 2.1. Bioinformatic Analysis of Fungal Rad6 Homologs

The standard *S. cerevisiae* Rad6 (NP_011457) served as the bait in search of the genomes of *B. bassiana* [22], as well as certain human and plant pathogenic fungi, and entomopathogenic fungi at http://blast.ncbi.nlm.nih.gov/Blast.cgi/ (accessed on 8 October 2019). A neighbor-joining method in MEGA7 was used to reveal the phylogenetic relationship between *B. bassiana* Ras6 and its orthologs [23].

### 2.2. Generation of rad6 Mutants

The WT strain of *B. bassiana* ARSEF 2860 (Bb2860) used in this study was obtained from the U.S. Department of Agriculture Entomopathogenic Fungus Collection (Ithaca, NY, USA). The *rad6* disruption strain, Δ*rad6*, was generated by homologous recombination in the wild-type *B. bassiana* strain ARSEF 2860. This process involved separating its coding/flanking fragments with a *bar* marker, a resistant gene against phosphinothricin. The complementary strain (Δ*rad6::rad6*) was designed by ectopic integration of a cassette containing the entire open reading frame (ORF) of the *rad6* gene and a *sur* marker, a resistant gene against chlorimuron ethyl. Briefly, the upstream (1541 bp) and downstream (1553 bp) fragments of the *rad6* ORF were amplified and ligated to the restriction enzyme sites *Eco*RI/*Bam*HI and *Xba*I/*Spe*I of the p0380-GFP-*bar* vector using the ClonExpress II One Step Cloning Kit (Vazyme Biotech, Nanjing, China). For the complementary strain, the promoter, coding sequence, and downstream regions were amplified and inserted into p0380-*sur*-gateway, resulting in p0380-*sur*-*rad6*. The disruption plasmid p0380-GFP-*rad6*-*bar* and complementation plasmid p0380-*sur*-*rad6* were transformed into the WT and Δ*rad6*, respectively, by using Agrobacterium-mediated transformation method, following the procedures detailed in our previous studies [1,24,25]. The putative disruption transformants were initially screened on CDA plates (200 μg/mL phosphinothricin or 15 μg/mL chlorimuron ethyl), followed by a secondary LSCM screening to identify non-fluorescent strains. Positive transformants were subjected to PCR verification. The primers used for generating the mutants are shown in Appendix A. The results of the verification of the mutants are described in our previous publication [26].

### 2.3. Cultivation Conditions

The strains were cultured in Sabouraud dextrose agar (SDAY) medium (4% glucose, 1% peptone, 1% yeast extract, and 1.5% agar) at 25 °C on a 12:12 h light/dark cycle for 8 days, and then the concentration of conidia was adjusted to 10^7^ conidia mL^−1^ for subsequent experiments. SDAY and the nutrient-poor medium, Czapek-Dox agar (CDA) (3% sucrose, 0.3% NaNO_3_, 0.1% K_2_HPO_4_, 0.05% KCl, 0.05% MgSO_4_, and 0.001% FeSO_4_, plus 1.5% agar), were used for phenotypic assay and the fungal culture for transcriptomic analysis.

### 2.4. Hyphal Biomass and Conidial Production

For hyphal biomass and conidial production experiments, 100 μL aliquots of a 10^7^ conidia mL^−1^ suspension were evenly spread onto the cellophane-covered SDAY and CDA plates, followed incubation at 25 °C with a 12/12 light/dark cycle. For the hyphal biomass test, samples from each strain (WT, Δ*rad6*, and Δ*rad6::rad6*) were collected and dried to a constant weight for hyphal biomass analysis. For conidiation evaluation, three 5 mm fungal plugs were collected from each plate and then subjected to ultrasonic vibration in 1 mL of 0.02% Tween 80 liquid to quantify conidiation capability. Samples were collected on days 4, 5, 6, and 7, with three independent repeats.

### 2.5. Transcriptome Analysis and Verification

As described above, cultures of WT and Δ*rad6* strains were collected after incubating at 25 °C with a 12/12 light/dark cycle for four days. For transcriptomic analysis, three independent samples from each strain were submitted to Vazyme Biotech Co. (Nanjing, China). Briefly, the mRNA was extracted from each culture using magnetic oligo(dT) beads and fragmenting into little segments. The segments and random hexamer primers were used as templates to synthesize the first-strand cDNA, which were further used for the second-strand cDNA synthesis. The double-strand cDNA was end-repaired by adding singular adenines to the ends. After being mixed with proper adaptors, the cDNA library was sequenced on an Illumina HiSeq™ platform in Majorbio (Shanghai, China). To identify differentially expressed genes (DEGs) between WTs and *rad6* mutant strains, the log_2_ R(Δ*rad6*/WT) was calculated. DEGs were defined as downregulated (log_2_ R(Δ*rad6*/WT) ≤ −1) or upregulated (log_2_ R(Δ*rad6*/WT) ≥ 1) at a significance level of q (corrected *p*) < 0.05. All DEGs were further analyzed for GO and Kyoto Encyclopedia of Genes and Genomes (KEGG) enrichment at http://vip.majorbio.com/ (accessed on 1 March 2024). The transcription level of key genes involved in the central development pathway was obtained from the date of SDAY and CDA transcriptomes (Appendix A. The real-time quantitative (qRT-PCR) analyses were performed using SYBR Green assay for transcriptome data verification. The WT and the *rad6* disruption strain were incubated under the same conditions described in Section 2.2, followed by RNA extraction. The one-step gDNA removal and DNA synthesis kit (TransGen Biotech, Beijing, China) was used for the reverse transcription and synthesis of the cDNA. Five downregulated and five upregulated genes from the transcriptome data were selected for qRT-PCR verification, and the primers were designed using the online tool Primer3 (https://bioinfo.ut.ee/primer3-0.4.0/) (accessed on 14 September 2023), followed by the primer specificity checked with NCBI Primer-Blast (https://www.ncbi.nlm.nih.gov/tools/primer-blast/, accessed on 14 September 2023) (Appendix A).

### 2.6. Phenotype Validation Experiments

For conidial production, 100 μL of conidial suspension at a concentration of 10^7^ conidia mL^−1^ was uniformly inoculated into SDAY and CDAY Petri plates containing varying concentrations of biotin. Biotin was filtered and sterilized through a 0.22 μm sterile filter (Membrane: PES, Syringe filter, Tianjin, China) and then added to SDAY and CDA culture media that had been sterilized at high temperatures, with final concentrations of 1.5 µM, 3 µM, and 6 µM, respectively. Three culture samples (5 mm plugs) were collected from each dish, and ultrasonic vibration was applied to calculate conidia yield.

### 2.7. Statistical Analysis

Data collected from three independent experimental replicates were subjected to one-factor (strain) analysis of variance, followed by Tukey’s honestly significant difference (HSD) test, to assess the variance in means between the Δ*rad6* strain and its control strains.

## 3. Results

### 3.1. Bioinformatical Analysis of Rad6 Protein in B. bassiana

The study conducted a bioinformatic analysis of the Rad6 protein in *B. bassiana*. Specifically, in the *B. bassiana*, BBA_01469 was identified as a homologous gene to *S. cerevisiae* Rad6 (NP_011457), with an identity of 77.5% and an E value of 2 × 10^−92^, named Rad6 herein. *rad6* is a gene with three introns and encodes a 151-amino acid protein (molecular mass: 17.2 kD; isoelectric point: 5.44). This protein includes a Ubiquitin-conjugating enzyme E2 domain (UBQ-conjugat_E2; residues 7–145) and two E3 interaction residues (residues 64–65; 98–99), separated by an active site cysteine at position 88 (Figure 1a). Notably, the typical domains and residues align precisely with Rad6 orthologs in *S. cerevisiae*, *Aspergillus nidullans*, and *Aspergillus flavus*. However, no nuclear localization signal was found, except for residues 9–19 in *S. cerevisiae*. In a comparative analysis of all fungal Rad6 homologous protein sequences, the phylogenetic tree revealed two branches: filamentous fungi and yeast. The identity of the filamentous fungal branch protein sequences exceeded 93%, while the yeast branch ranged from 77% to 84%. The filamentous fungal branch could be roughly subdivided into two smaller branches: pathogenic fungi (with a higher similarity ranging from 96% to 100%) and non-pathogenic *Aspergillus* and *Penicillium* (with a lower similarity ranging from 93% to 95%) (Figure 1b).

### 3.2. Vital Role of Rad6 Protein in Conidiation and Hyphal Growth

The Δ*rad6* strains exhibited significant defects in both conidial production and hyphal growth, irrespective of whether they were cultured in nutrient-rich or nutrient-poor media. When cultured on SDAY plates, the conidial production of Δ*rad6* was significantly dropped by approximately 94% after a 7-day culture (Figure 2a). The dry hyphal mass of the Δ*rad6* was also significantly lower than that of the control strains (Figure 2b). Similarly, on nutrient-poor CDA plates, the Δ*rad6* strains produced only 33.65% of the conidia that the WT strains did after four days of incubation and reached 33.78% after seven days (Figure 2c). The dry hyphal biomass of Rad6 disruption mutant ranged from 40.93% to 54.22% of that in WT strains (Figure 2d). Generally, the phenotypic defects caused by disruption of *rad6* could be rescued by its complementation. The dry hyphal biomass and the conidial production of Δ*rad6::rad6* cultured on SDAY or CDA were restored to a level close to that of WT.

### 3.3. Transcriptomic View of Rad6-Mediated Regulation Pathway Differences

Transcriptomic analysis was carried out to reveal the gene expression patten differences between Δ*rad6* and WT, cultured on both the nutrient-rich medium SDAY and nutrient-poor medium CDA. The volcano plot exhibits a strikingly similar pattern in the disruption and WT strains incubated in both SDAY and CDA media. Specifically, 1976 downregulated and 2109 upregulated differentially expressed genes (DEGs) were identified in Δ*rad6* vs. WT strains incubated in SDAY, while 1437 and 1794 DEGs were found in CDA cultures (Figure 3a,b). KEEG enrichment analysis indicated that the disruption strains exhibited alterations in the DNA replication, tryptophan, and biotin metabolism pathways in both transcriptomes (Figure 3c,d).

Further investigation revealed the existence of 2006 total DEGs, comprising 939 upregulated and 813 downregulated DEGs, common to both transcriptomes (Figure 4a–c). Enrichment analysis of these common DEGs identified DNA replication and the biotin metabolism pathway as the top two significantly influenced pathways (Figure 4d). Interestingly, the transcriptional levels of genes *flbC*, *brlA*, *abaA*, and *wetA* were significantly downregulated in both nutrient-rich/poor transcriptomes (Appendix A). In *B. bassiana*, the function of *flbC*, *brlA*, *abaA*, and *wetA* was characterized as critical genes in the central developmental pathway (CDP), which affects the fungal asexual development [13,14,15]. These results show that *rad6* may be involved in fungal conidiation by regulating the transcription of various genes in *B. bassiana*.

### 3.4. Conidiation Increases with Biotin Addition

To investigate the potential biotin metabolism disorder, conidia from Δ*rad6* strains and the control strains were inoculated onto SDAY and CDA plates containing various biotin concentrations and incubated for eight days under optimal conditions. On the SDAY plates, the conidial production of Δ*rad6* was 6.13 × 10^7^ conidia per cm^2^, only 9.52% of that of WT. However, it showed a significant increase (*p*-value < 0.05) to 39.24 × 10^7^ conidia per cm^2^ after the addition of 6 µM biotin (Figure 5a). The relative conidiation inhibition (RCI) rates due to the addition of biotin were significantly lower in Δ*rad6* than that in WT (Figure 5b). On CDA plates, the conidial production of Δ*rad6* was 0.89 × 10^7^ conidia per cm^2^, 33.78% of that of the WT strains. However, it showed a significant increase (*p*-value < 0.05) to 1.03 × 10^7^ conidia per cm^2^ after the addition of 6 µM biotin (Figure 5c). The relative conidiation inhibition (RCI) rates due to the addition of biotin were also significantly lower in Δ*rad6* strains than that in WT (Figure 5d).

### 3.5. Transcription Pattern of CDP Genes in Different Media with and without Exogenous Biotin

In the transcriptomic validation experiment, all ten genes that exhibited downregulation in the SDAY and CDA transcriptomics data were consistently and significantly downregulated in qRT-PCR. Similarly, all ten genes that showed upregulation in the transcriptomics data displayed significant upregulation in the qRT-PCR results (Appendix A). A noteworthy observation emerged when assessing the expression levels of key genes involved in the central developmental pathway. Compared to strains cultivated on SDAY without the addition of biotin, those exposed to biotin exhibited a substantial increase in the expression of key genes associated with this central development pathway (Figure 6a,b). This finding strongly suggests that biotin enhances the expression of pivotal genes within the central development pathway, consequently bolstering conidial production. In the case of CDA, the transcription levels of *brlA*, *abaA*, *wetA*, and *vosA* were 51.76%, 62.27%, 54.60%, and 220.12% of that in WT (Figure 6c). When exogenous biotin was added, their transcription levels were 47.39%, 119.35%, 170.82%, and 286.60% of that in WT (Figure 6d). This observation suggests that biotin augments conidial production by stimulating the expression of *abaA*, *wetA*, and *vosA*. These results consistently support our prior analysis.

## 4. Discussion

Post-modification ubiquitination profoundly influences various aspects of cellular events in model organisms such as *S. cerevisiae*, *Arabidopsis thaliana*, and *Homo sapiens* [27,28], thereby exerting a significant influence on the fate of life across kingdoms. In this study, we investigated the function of Rad6 in *B. bassiana*, a widely used filamentous entomopathogenic fungus for biocontrol. Our experimental data demonstrated that the *rad6* gene affects the conidial production capacity of *B. bassiana* under both nutrient-rich and nutrient-poor conditions. Interestingly, we observed that, in addition to being regulated by the central developmental pathway, the conidiation deficiency in Δ*rad6* can be partially alleviated through the exogenous addition of biotin, as discussed below.

The deletion of *rad6* leads to a dramatic loss in conidial production in both SDAY and CDA incubated strains, which is consistent with the previous studies in *S. cerevisiae* and *M. oryzae* [17,29]. The deletion of Rad6 reduced about 67% of the conidiation potential but only 45% of the hyphal mass in poor CDA plates. Similar but slightly different, in rich SDAY media, the absence of Rad6 caused a 94% conidia yield decrease but 55% hyphal mass loss. The higher percentage of conidial production rate loss and lower hyphal mass rate loss of Δ*rad6* compared to that in WT indicated that the absence of Rad6 plays a more substantial role in inhibiting conidial production than hyphal growth.

In filamentous fungi *Aspergillus* and *Penicillium*, the strain conidial production is usually mediated by the central development pathway consisting of the activators of BrlA, AbaA, and WetA [30]. The deletion of BrlA and AbaA has no influence on hyphal growth but abolishes aerial conidiation in *B. bassiana* [13]. WetA contributes to nearly 98% of the conidiation capacities of *B. bassiana* [14]. The downregulated transcription level of CDP-related genes in Δ*rad6* grown on SDAY and CDA explained its dramatic conidiation decrease. VosA is a downstream gene of CDP, which can also contribute to 88% of the conidiation capacities in *B. bassiana* [14]. The significant downregulation of VosA in SDAY cultures and upregulation in CDA cultures may have caused the profound conidiation loss in the former. On the other hand, the more severe down-regulation of CDP genes in SDAY cultures could be another reason. Fluffy genes *flbA–flbE* are well-known activators of *brlA* in *Aspergillus nidulans* [30,31]. Among them, flbA significantly negatively regulated the hyphal biomass of *B. bassiana* [15]. Therefore, the up-/down-regulated transcriptional level of *flbA* in Δ*rad6* cultured on SDAY/CDA, respectively, may provide an important insight into the more severe hyphal mass defect observed in Δ*rad6* when cultured on SDAY compared to CDA.

The KEGG analysis based on the transcriptomic data of Δ*rad6* vs. WT revealed that biotin metabolism abnormality is one of the key reasons for conidial reproduction. Biotin, a water-soluble vitamin, is indispensable for the viability of a broad spectrum of organisms, from bacteria to eukaryotes [32]. In filamentous fungi, biotin enhances the colonization and proliferation capability of *Candida albicans* in the catheter lumen [33]; it is an essential growth factor for the pathogenic fungus *Allescheria boydii* [34]. Adding exogenous biotin to the minimal medium could not only increase the formation of wild-type conidias by nearly three-fold, but also alleviate the conidial formation defect in the bioB knockout strain of *Alternaria alternata* [8]. In this study, with the addition of 6 µM biotin, the conidia yield of Δ*rad6* significantly increased by 45.64% and 48.77% when cultured on SDAY and CDA media. These results verify the hypothesis that the exogenous addition of biotin benefits conidial production in *B. bassiana*, which is consistent with the results that biotin can restore the growth defects in *Aspergillus oryzae* [35]. Apart from being a covalently bound coenzyme for carboxylases, evidence has shown that biotin can be attached to histones [36] and plays a role in gene repression, heterochromatin structures, and repression of transposons [37,38]. We suggest that the conidial yield reduction is caused by downregulated CDP genes whose transcription level may be indirectly influenced by the biotin metabolism disorder. This hypothesis gains substantial support from the significant upregulation of CDP-related genes in Δ*rad6* cultured with biotin compared to that in Δ*rad6* cultured without biotin.

In summary, Rad6 profoundly influences the conidiation capacity of *B. bassiana* under nutrient-rich and nutrient-poor conditions. Transcriptomic analysis of cultures grown in these diverse nutritional environments indicates a pivotal role of biotin in modulating conidial production, a finding substantiated through biotin supplementation. Further exploration suggests that biotin may impact conidial generation via histone biotinylation, consequently affecting gene transcription within the central developmental pathway. While these discoveries underscore the significant involvement of Rad6 in fungal asexual development, the precise mechanisms by which Rad6 influences biotin metabolism through ubiquitination and how biotin suppresses the expression of CDP-related genes remain elusive, warranting further investigation in subsequent studies.

## Figures and Tables

**Figure 1 jof-10-00613-f001:**
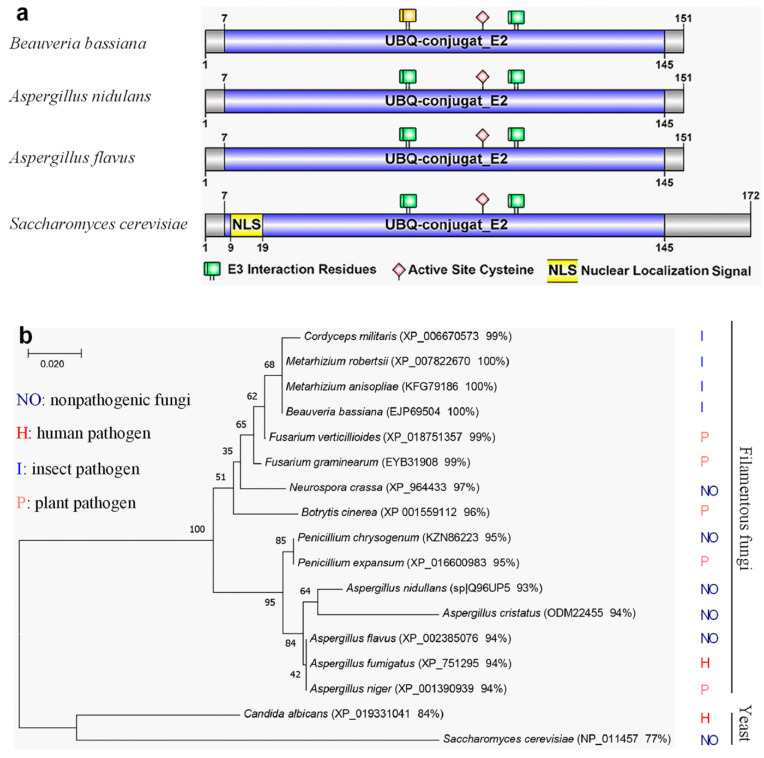
Structural analysis of *B. bassiana* Rad6 and its phylogenetic relationship with other fungal homologues. (**a**) Domain analysis of *B. bassiana* Rad6. (**b**) The phylogenetic relationship between *B. basiana* Rad6 and other fungal homologues. Bootstrap values obtained through 1000 replications are presented at the nodes for reference. The branch length in the scale is proportional to the genetic distance, as determined using the neighbor-joining method within the MEGA7 program, accessible at http://www.megasoftware.net/ (accessed on 8 October 2019) (note: the accession number of each protein in NCBI and its similarity to the corresponding protein of *Beauveria bassiana* Rad6 are shown in parentheses).

**Figure 2 jof-10-00613-f002:**
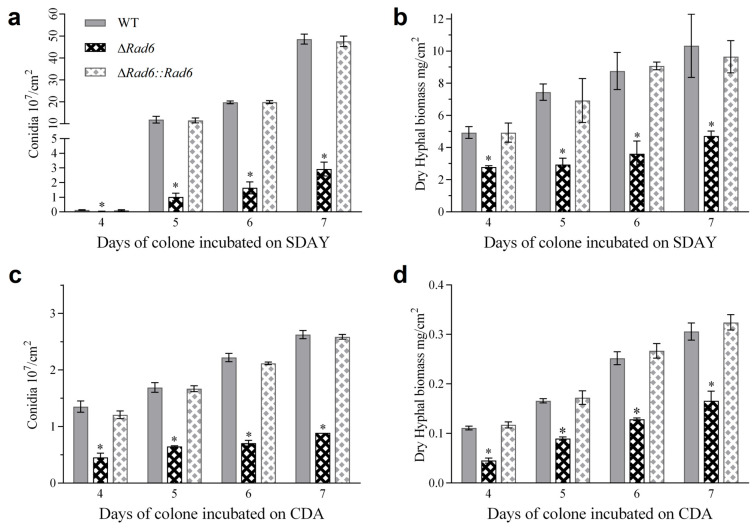
Impacts of Rad6 disruption on conidiation capability and hyphal growth of *B. bassiana* in different culture media. (**a**) Conidia yield of each strain cultured in nutrient-rich SDAY plates. (**b**) Dry hyphal biomass of strains collected from nutrient-rich SDAY plates. (**c**) Conidia yield of each strain cultured in nutrient-poor CDA plates. (**d**) Dry hyphal biomass of strains collected from nutrient-poor CDA plates. All experiments were initiated by spreading 100 μL of a 10^7^ conidia mL^−1^ suspension. Asterisked bars in each group significantly differ from unmarked bars (Tukey’s HSD, *p* < 0.05). Error bar: SD from three replicates.

**Figure 3 jof-10-00613-f003:**
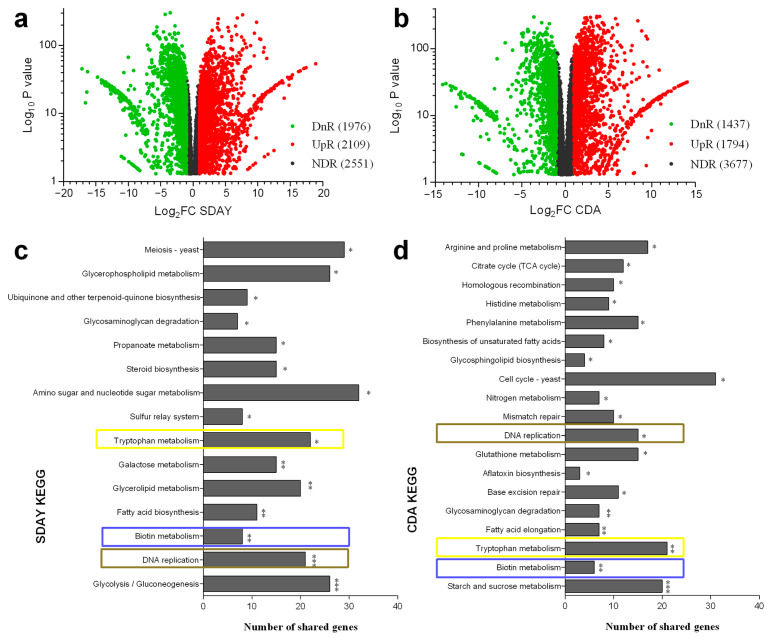
Transcriptome analysis of Δ*rad6* vs. WT collected from different culture media. (**a**,**b**) Distribution of log_2_ FC and *p* values for genes identified in the transcriptomes from SDAY and CDA cultures. (**c**,**d**) KEGG pathway enrichment analysis of DEGs from SDAY and CDA cultures. Differentially expressed genes (DEGs) are defined as log_2_ FC ≤ –1 or log_2_ FC ≥ 1 at the level of *p* < 0.05. The remaining genes are insignificantly affected (–1 ≤ log_2_ FC ≤ 1). (Note: DnR refers to downregulated, UpR refers to upregulated, NDR refers to non-significant differentially regulated. The pathways are ordered based on the significant results in the KEGG enrichment analysis. *p*-value < 0.001 labeled ***, *p*-value < 0.01 labeled **, and *p*-value < 0.05 labeled *).

**Figure 4 jof-10-00613-f004:**
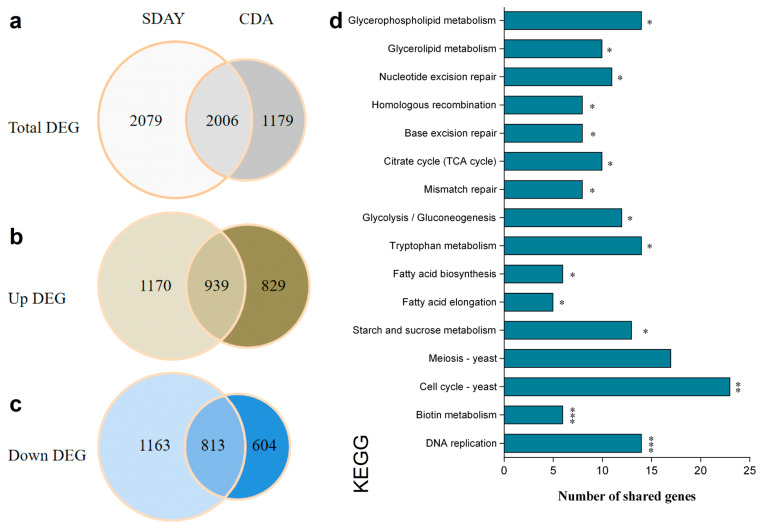
Analysis of genes co-occurring in the transcriptome of different media in *B. bassiana*. (**a**–**c**) Overlapped in the total, upregulated, and downregulated DEGs from SDAY and CDA cultures in *B. bassiana*. (**d**) KEGG pathway enrichment analysis of common DEGs from SDAY and CDA cultures. The pathways are ordered based on the significant results in the KEGG enrichment analysis. *p*-value < 0.001 labeled ***, *p*-value < 0.01 labeled **, and *p*-value < 0.05 labeled *.

**Figure 5 jof-10-00613-f005:**
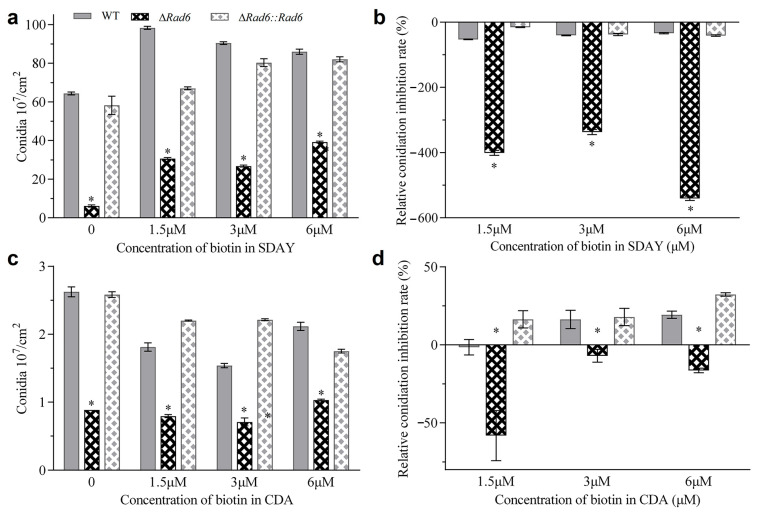
Phenotype verification of biotin addition. (**a**,**c**) Conidiation of each strain in SDAY and CDA plates amended with different concentrations of biotin. (**b**,**d**) Relative conidiation inhibition rate of biotin addition at different concentrations. Asterisked bars in each group significantly differ from unmarked bars (Tukey’s HSD, *p* < 0.05). Error bar: SD from three replicates.

**Figure 6 jof-10-00613-f006:**
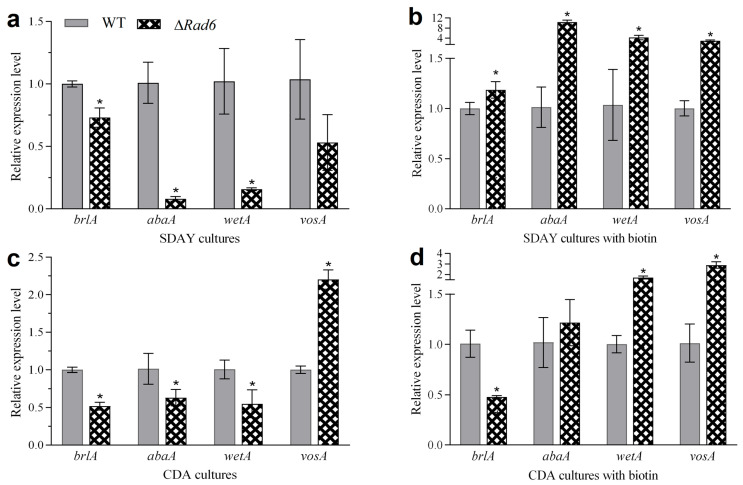
qRT-PCR verification with or without exogenous biotin. (**a**,**c**) Transcription levels of CDP-related genes in biotin-free SDAY and CDA plates. (**b**,**d**) Transcription levels of CDP-related genes in SDAY and CDA plates with biotin added. Asterisked bars in each group significantly differ from unmarked bars (Tukey’s HSD, *p* < 0.05). Error bar: SD from three replicates.

## Data Availability

All data generated or analyzed during this study are included in this published article (and its Appendix A). All engineered strains used in this study are available upon reasonable request. All data generated or analyzed during this study are deposited in figshare, DOI: https://doi.org/10.6084/m9.figshare.26156842. The transcriptome sequencing data were deposited in the NCBI database with the accession number PRJNA1037315.

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
