# Peer review of "Rad6 Regulates Conidiation by Affecting the Biotin Metabolism in Beauveria bassiana"

_jof, 2024, doi:10.3390/jof10090613_

Round 1

Reviewer 1 Report

The work provides relevant information on the role of the rad6 gene in sporulation. However, I consider that it could be a scientific note as the study was quite limited. The authors could provide more background on how the muntante was carried out, it would give more support to the work.

Indicate all my comments in the previous section

Author Response

We appreciate you very much for your positive and constructive comments on our manuscript. We have provided a detailed response to your conments. Please review the uploaded file. Meanwhile, we conduct comprehensive language editing on the manuscript to improve its readability. The modified parts in the manuscript are marked in red, for your kind consideration.

Reviewer 1 Comments

Point 1:The authors should incorporate the reference of the publication describing how the deletion and complementation of the rad6 gene in the ARSEF 2860 strain was carried out and corroborated. It would have provided more support for this research if the rad6 gene deletion and complementation methodology had been performed in this work.

Response:Thank you so much for your comment. The relevant information has been added in the revised manuscript in Line 98: “The rad6 disruption strain, Δrad6, was generated by homogeneous recombination in the wild-type B. bassiana strain ARSEF 2860. This process involved separating its coding/flanking fragments with a bar marker, a resistant gene against phosphinothricin. The complementary strain (Δrad6::rad6) was designed by ectopic integration of a cassette containing the entire open reading frame (ORF) of the rad6 gene and a sur marker, a resistant gene against chlorimuron ethyl. Briefly, the upstream (1541bp) and downstream (1553bp) fragments of the rad6 ORF were amplified and ligated to the restriction enzyme sites EcoRI/BamHI and XbaI/SpeI of the p0380-GFP-bar vector using the ClonExpress II One Step Cloning Kit (Vazyme Biotech, Nanjing, China). For the complementary strain, the promoter, coding sequence, and downstream regions were amplified and inserted into p0380-sur-gateway, resulting in p0380-sur-rad6. The disruption plasmid p0380-GFP-rad6-bar and complementation plasmid p0380-sur-rad6 were transformed into the WT and Δrad6, respectively, by using Agrobacterium-mediated transformation method, following the procedures detailed in our previous studies [24,25,26]. The putative disruption transformants were initially screened on CDA plates (200 μg/ml phosphinothricin or 15 μg/ml chlorimuron ethyl), followed by a secondary LSCM screening to identify non-fluorescent strains. Positive transformants were subjected to PCR verification. The primers used for generating the mutants were shown in Supplemental file 1, Table S1. The results of the verification of the mutants were described in our previous publication [27].”

Point 2:CDP genes are mentioned several times in the text but the role they play is not explained.

Response:Thank you so much for your comment. In the resubmitted manuscript, we have supplemented more information on the CDP pathway and CDP genes on Line 61: “In filamentous fungi, the central developmental pathway (CDP) plays a pivotal role in influencing fungal asexual development by regulating the expression of downstream conidiation-specific genes through cascade regulation. BrlA, AbaA, and WetA are three key regulators in the CDP [11,12]. In B. bassiana, disruption of brlA and abaA not only results in the elimination of conidia and blastospore production, but also inhibits its dimorphic transition [13]. WetA and its downstream velvet protein, VosA, are associated with conidial maturation [14]. In addition, flb genes, including flbA-E, are essential for CDP activation by mediating the transcription of brlA and abaA [15]. These findings indicate that the CDP has a complex regulatory network and is indispensable in the asexual development of B. bassiana.

Point 3:In figure 1 b indicate what the value in brackets refers to.

Response:Thank you so much for your comment. In the resubmitted manuscript, We have added relevant information on Line 194: “ (Note: The accession number of each protein in NCBI and its similarity to the corresponding protein of Beauveria bassiana Rad6 are shown in parentheses ).”

Point 4:Figure 2b does not present the results obtained for dry hyphal biomass of the wild type strain.

Response:Thank you so much for your comment. In the resubmitted manuscript, we have added the results for dry hyphal biomass of the wild type strain in figure 2d (We have adjusted the original Figure 2b to the current position of Figure 2d).

Point 5:In section 3.1 it should be indicated against which Ra6 gene high homology (77.5% identity) was obtained. Is it the NP_011457 gene?

Response:Thank you so much for your comment. In the resubmitted manuscript, we have made modifications to Line 172: “Specifically, in the B. bassiana, BBA_01469 was identified as a homologous gene to S. cerevisiae Rad6 (NP_011457), with an identity of 77.5% and an E value of 2e-92, named Rad6 herein.”

Point 6:In section 3.2, there is absolutely no comment of the ARSEF 2860 deletion rad6::rad6 strain.

Response:Thank you so much for your comment. In the resubmitted manuscript, we have added the relevant information to Line 206: “Generally, the phenotypic defects caused by disruption of rad6 could be rescued by its complementation. The dry hyphal biomass and the conidial production of Δrad6::rad6 cultured on SDAY or CDA were restored to a level close to that of WT.”

Point 7:Figure 3 should specify that the transcriptome analysis was carried out between the mutant and WT strain.

Response:Thank you so much for your comment. We are grateful for your professional suggestions. In the resubmitted manuscript, the relevant information has been added in Line 219: “Transcriptomic analysis was carried out to reveal the gene expression patten differences between Δrad6 and WT, cultured on both nutrient-rich medium SDAY and nutrien-poor medium, CDA.”

Point 8:The figures in the paper should below each result. This way you have a context of what the figure is about.

Response:Thank you so much for your comment. We are grateful for your professional suggestions. In the resubmitted manuscript, we have adjusted the corresponding figures below each result.

Point 9:Section 3.3 should indicate the function of the critical genes, flbC, brlA, abaA and wetA in the text or add a table in the main text of the manuscript.

Response:Thank you so much for your comment. In the resubmitted manuscript, we have added new content on Line 234: “In B. bassiana, the function of flbC, brlA, abaA, and wetA has been characterized as critical gens in the central developmental pathway (CDP) which affects the fungal asexual development [13,14,15]. These results showed that rad6 may be involved in fungal conidiation by regulating the transcription of various genes in B. bassiana.”

Point 10:In Figure 6, the expression of the genes of interest in the complemented strain could have been evaluated.

Response:Thank you so much for your comment. We appreciate your valuable suggestions. However, since the transcriptome analysis in our article is based on WT and rad6 knockout strains, we decided to show the expression levels of key genes related to the Central developmental pathway (CDP) in WT and knockout strains in Figure 6. This result can also be mutually validated with the transcriptomic results, increasing the reliability of the experiment.

Point 11: In the discussion section, reference should be made to the figures when discussing a result that was plotted.

Response:Thank you so much for your comment. However, in our opinion, clarification of the results based on figures might be completed in the “Results” section. Thank you for your understanding.

Point 12: The paragraph ‘The more severe....’ (lines 267-269) could be better formulated (lines 267-269).

Response:Thank you so much for your comment. In the resubmitted manuscript, we have revised the description in Line 322: “Therefore, the up-/down-regulated transcriptional level of flbA in Δrad6 cultured on SDAY/CDA, respectively, may provide an important insight into the more severe hyphal mass defect observed in Δrad6 when cultured on SDAY compared to CDA.”

Reviewer 2 Report

The article demonstrates that rad6 mutants in Beauveria bassiana decrease conidial production and that this conidial and biomass production can be partially restored by the addition of biotin. This highlights the importance of rad6 in biotin metabolism in entomopathogenic fungi, as well as in the production of conidia and biomass in Beauveria bassiana. This is relevant because Beauveria bassiana is one of the most important and commercially used microbial biocontrol agents, whose infective unit is the conidium.

Line 12: Beauveria bassiana (not in italics).  Necessary

Line 24: The acronym CDP should be defined (does it refer to central developmental pathway?). Necessary

Line 50: It is mentioned that in recent years "there has also been evidence of novel roles for biotin in cell signaling, gene expression, and chromatin structure" [5,6,7]. The citations are from 2003, 2005, and 2019. Desirable

Line 56-57: Candida is a genus, it should be italicized. Necessary

Line 61: rad6 is mentioned as a gene, it should be italicized.  Desirable

Line 83: The hyperlink does not work, it should be cited as proposed by the website (Kumar et al. 2016). Desirable

Line 84: The acronym WT has already been defined on line 75, it is not necessary to clarify it again.

Line 90-91: Clarify that these are not the standard concentrations of the Sabouraud dextrose agar (SDAY) medium; this is mentioned in line 100. Desirable

Line 125: It does not mention what type of qPCR was used; from the supplementary material, I infer that it was Sybr Green. It is not clear whether the oligonucleotides were custom-designed or where they were obtained from, nor their efficiency. Necessary

Throughout the text, the terms spore and conidia are used; it should be homogeneous as conidia, unless each term is defined. Necessary

Line 129: Mention the concentrations of biotin used, how they were sterilized, and if they were added after the sterilization of the medium.  Desirable

Line 146-147: To avoid confusion, I propose: "Bbrad6 is a gene with three introns and encodes a 151-amino acid protein." Desirable

Line 158: Aspergillus and Penicillium should be italicized as they are scientific names. Necessary

Figure 2b: This part is very important, influencing the results and discussion; the dry biomass weight bar of WT does not appear at any time (results mentioned in lines 172 and 173). Necessary

Figure 2: It does not mention what the asterisks or error bars (SD or SE) mean, nor the significance (P=>< #) of the statistical analysis. Necessary

Figure 3: It should clarify in the DEGs between which strains the comparisons are made (RAD6 mutant, WT), and also mention why the gene list (by function) is not in the same order between those grown in SDAY and CDA. What is the criterion for ordering them? It does not mention what the asterisks mean nor the significance (P=>< #) of the statistical analysis. Necessary

Figure 4: Same clarification as in Figure 3. Necessary

Line 194 and 196: Same clarification as in Figure 3 and Figure 4. Necessary

Figure 5: Same comments as in Figure 2. Necessary

Lines 210-213: Spore production is mentioned as a percentage citing Figure 5a, which is in the number of conidia per ml.   Desirable

Figure 6: Review the figure legend text; the word "without" is repeated. Take into account the suggestions for the other figures. Necessary

Line 230-233: It is mentioned that there is no impact on brlA, but the least impact is seen in abaA (which does not have an asterisk). It is important to clarify this part as it modifies the discussion.  Necessary

Lines 245-246: "The deficiency in conidiation can be partially alleviated through the exogenous addition of biotin, as discussed below." This statement is not statistically supported. Necessary

Line 280: Use a more objective term than "believed", such as "we suggest" or "we propose". Desirable

Author Response

We appreciate you very much for your positive and constructive comments on our manuscript. We have provided a detailed response to your conments. Please review the uploaded file. Meanwhile, we conduct comprehensive language editing on the manuscript to improve its readability. The modified parts in the manuscript are marked in red, for your kind consideration.

Reviewer 2 Comments

Point 1:Most of the methodology is clear and concise; however, it could further detail the validation of the transcriptional analysis by qPCR.

Response:Thank you so much for your comment. We have added the relevant information in Line 149 in the resubmitted manuscript: “The real-time quantitative (qRT-PCR) analyses were performed using SYBR Green assay for transcriptome data verification. WT and the rad6 disruption strain were incubated under the same conditions described in 2.2, followed by RNA extraction. The one-step gDNA removal and DNA synthesis kit (TransGen Biotech Beijing, China) was used for the reverse transcription and synthesis of the cDNA. Five downregulated and five upregulated genes from the transcriptome data were selected for qRT-PCR verification and the primers were designed using the online tool Primer3 (https://bioinfo.ut.ee/primer3-0.4.0/), followed by the primer specificity checked with NCBI Primer-Blast (https://www.ncbi.nlm.nih.gov/ tools/primer-blast/) (Supplemental file 1 Table S2).”

Point 2:For the most part, the results are clear; however, the level of significance is not mentioned, and the graphs do not define what the asterisks or error bars represent. The arrangement of the graphs should be consistent across all figures. In some, the SDAY results appear on the left and the CDA results on the right; in others, it is reversed, and in others, they are arranged top and bottom.

Response:Thank you so much for your comment. In the resubmitted manuscript, We have rearranged the chart and added relevant information of asterisks and error bars in the caption.

Point 3:Line 12: Beauveria bassiana (not in italics).

Response:Thank you so much for your comment. We are grateful for your professional suggestions to help us improve this manuscript. This clerical error has been revised in the resubmitted manuscript.

Point 4:Line 24: The acronym CDP should be defined (does it refer to central developmental pathway?).

Response:Thank you so much for your comment. In the resubmitted manuscript, we introduced CDP in the introduction section in Line 61: “In filamentous fungi, the central developmental pathway (CDP) plays a pivotal role in influencing fungal asexual development by regulating the expression of downstream conidiation-specific genes through cascade regulation. BrlA, AbaA, and WetA are three key regulators in the CDP [11,12]. In B. bassiana, disruption of brlA and abaA not only results in the elimination of conidia and blastospore production, but also inhibits its dimorphic transition [13]. WetA and its downstream velvet protein, VosA, are associated with conidial maturation [14]. In addition, flb genes, including flbA-E, are essential for CDP activation by mediating the transcription of brlA and abaA [15]. These findings indicate that the CDP has a complex regulatory network and is indispensable in the asexual development of B. bassiana.

Point 5:Line 50: It is mentioned that in recent years "there has also been evidence of novel roles for biotin in cell signaling, gene expression, and chromatin structure" [5,6,7]. The citations are from 2003, 2005, and 2019.

Response:Thank you so much for your comment. We have revised the description at Line 48: “As one of the vital vitamins for all living organisms, biotin plays a crucial role in many important biological processes, such as biotin-dependent carboxylases, fatty acid biosynthesis, gluconeogenesis and amino acid metabolism, cell signaling, gene expression, and chromatin structure [4,5,6].”

Point 6:Line 56-57: Candida is a genus, it should be italicized.

Response:Thank you so much for your comment. We are grateful for your professional suggestions to help us improve this manuscript. This clerical error has been revised in the resubmitted manuscript.

Point 7:Line 61: rad6 is mentioned as a gene, it should be italicized.

Response:Thank you very much for your comment. After reviewing the entire text, we confirmed that "rad6" is italicized throughout the entire text. We are grateful for your professional suggestions to help us improve this manuscript.

Point 8:Line 83: The hyperlink does not work, it should be cited as proposed by the website (Kumar et al. 2016).

Response:Thank you so much for your comment. We have revised the description in the resubmitted manuscript: “A neighbor-joining method in MEGA7 was used to reveal the phylogenetic relationship of B. bassiana Ras6 and its orthologs [23].”

Point 9:Line 84: The acronym WT has already been defined on line 75, it is not necessary to clarify it again.

Response:Thank you so much for your comment. We are grateful for your professional suggestions to help us improve this manuscript. This clerical error has been revised in the resubmitted manuscript.

Point 10:Line 90-91: Clarify that these are not the standard concentrations of the Sabouraud dextrose agar (SDAY) medium; this is mentioned in line 100.

Response:Thank you so much for your comment. The SDAY and CDA media used in our work are prepared with the standard concentration. To improve the readability of the literature, we revised the description in the resubmitted manuscript in Line 125: “For hyphal biomass and conidial production experiments, 100 μL aliquots of a 107 conidia/mL suspension were evenly spread onto the cellophane-covered SDAY and CDA plates, followed by incubated at 25°C with a 12/12 light/dark cycle.”

Point 11:Line 125: It does not mention what type of qPCR was used; from the supplementary material, I infer that it was Sybr Green. It is not clear whether the oligonucleotides were custom-designed or where they were obtained from, nor their efficiency.

Response:Thank you so much for your comment. We have added the relevant information in the section 2.4. Line 149: “The real-time quantitative (qRT-PCR) analyses were performed using SYBR Green assay for transcriptome data verification. ” Line 153: “Five downregulated and five upregulated genes from the transcriptome data were selected for qRT-PCR verification and the primers were designed using the online tool Primer3 (https://bioinfo.ut.ee/primer3-0.4.0/), followed by the primer specificity checked with NCBI Primer-Blast (https://www.ncbi.nlm.nih.gov/ tools/primer-blast/) (Supplemental file 1 Table S2).”

Point 12:Throughout the text, the terms spore and conidia are used; it should be homogeneous as conidia, unless each term is defined.

Response:Thank you so much for your comment. The “spore” and “conidia” used in our manuscript both refer to asexual germ cells produced by fungi. We have homogeneous them as “conidia” in the revised manuscript.

Point 13:Line 129: Mention the concentrations of biotin used, how they were sterilized, and if they were added after the sterilization of the medium.

Response:Thank you so much for your comment. We will add the concentration of biotin and the method of sterilization treatment to Line 161: “Biotin was filtered and sterilized through a 0.22 μm sterile filter (Membrane: PES, Syringe filter, Tianjin, China), and then added to SDAY and CDA culture media that have been sterilized at high temperatures, with final concentrations of 1.5 µM, 3 µM, and 6 µM, respectively.”

Point 14:Line 146-147: To avoid confusion, I propose: "Bbrad6 is a gene with three introns and encodes a 151-amino acid protein."

Response:Thank you very much for your comment. We are grateful for your professional suggestions. In the resubmitted manuscript, we have revised the information in Line 175: “rad6 is a gene with three introns and encodes a 151-amino acid protein (molecular mass: 17.2 kD; isoelectric point: 5.44).”

Point 15:Line 158: Aspergillus and Penicillium should be italicized as they are scientific names.

Response:Thank you so much for your comment. We are grateful for your professional suggestions to help us improve this manuscript. This clerical error has been revised in the resubmitted manuscript.

Point 16:Figure 2b: This part is very important, influencing the results and discussion; the dry biomass weight bar of WT does not appear at any time (results mentioned in lines 172 and 173).

Response:Thank you so much for your comment. In the resubmitted manuscript, we have added the results of dry hyphal biomass of the wild type strain.

Point 17:Figure 2: It does not mention what the asterisks or error bars (SD or SE) mean, nor the significance (P=>< #) of the statistical analysis.

Response:Thank you so much for your comment. We are grateful for your professional suggestions. In the resubmitted manuscript, we have added the relevant information to the captions of Figure 2, Figure 5, and Figure 6: “Asterisked bars in each group significantly differ from unmarked bars (Tukey’s HSD, p < 0.05). Error bar: SD from three replicates.”

Point 18:Figure 3: It should clarify in the DEGs between which strains the comparisons are made (RAD6 mutant, WT), and also mention why the gene list (by function) is not in the same order between those grown in SDAY and CDA. What is the criterion for ordering them? It does not mention what the asterisks mean nor the significance (P=>< #) of the statistical analysis.

Response:Thank you so much for your comment. Thank you so much for your comment. The relevant information has been added to the caption of Figure 3: “(Note: DnR refers to downregulated, UpR refers to upregulated, NDR refers to non-significant differentially regulated. The pathways were ordered based on the significant results in the KEGG enrichment analysis. p-value< 0.001 labelled ***, p-value< 0.01 labelled **, and p-value< 0.05 labelled *)”

Point 19:Figure 4: Same clarification as in Figure 3.

Response:Thank you so much for your comment. The relevant information has been added to the caption of Figure 4: “The pathways were ordered based on the significant results in the KEGG enrichment analysis. p-value< 0.001 labelled ***, p-value< 0.01 labelled **, and p-value< 0.05 labelled *.”

Point 20:Line 194 and 196: Same clarification as in Figure 3 and Figure 4.

Response:Thank you so much for your comment. The relevant information has been added in this section.

Point 21:Figure 5: Same comments as in Figure 2.

Response:Thank you so much for your comment. The relevant information has been added in the figure legend.

Point 22:Lines 210-213: Spore production is mentioned as a percentage citing Figure 5a, which is in the number of conidia per ml.

Response:Thank you so much for your comment. We have revised the description in Line 256: “On SDAY plates, the conidial production of Δrad6 was 6.13 × 107 conidia per cm2, only 9.52% of that of WT. However, it showed a significant increase (p-value < 0.05) to 39.24 × 107 conidia per cm2 after adding 6 µM biotin (Fig. 5a). The relative conidiation inhibition (RCI ) rates due to the addition of biotin were significantly lower in Δrad6 than that in WT (Fig. 5b). On CDA plates, the conidial production of Δrad6 was 0.89 × 107 conidia per cm2, 33.78% of that of WT strains. However, it showed a significant increase (p-value < 0.05) to 1.03 × 107 conidia per cm2 after adding 6 µM biotin (Fig. 5c). The relative conidiation inhibition (RCI ) rates due to the addition of biotin were also significantly lower in Δrad6 strains than that in WT (Fig. 5d). 

Point 23:Figure 6: Review the figure legend text; the word "without" is repeated. Take into account the suggestions for the other figures.

Response:Thank you so much for your comment. This clerical error has been revised in the resubmitted manuscript.

Point 24:Line 230-233: It is mentioned that there is no impact on brlA, but the least impact is seen in abaA (which does not have an asterisk). It is important to clarify this part as it modifies the discussion.

Response:Thank you so much for your comment. To improve the readability of our manuscript, we revised the description in Line 281: “In the case of CDA, the transcription levels of brlA, abaA, wetA, and vosA were 51.76%, 62.27%, 54.60%, and 220.12% of that in WT (Fig. 6c). When exogenous biotin addition, their transcription levels were 47.39%, 119.35%, 170.82%, and 286.60% of that in WT (Fig. 6d).”

Point 25:Lines 245-246: "The deficiency in conidiation can be partially alleviated through the exogenous addition of biotin, as discussed below." This statement is not statistically supported.

Response:Thank you so much for your comment. In the resubmitted manuscript, the statistically analysis has been added in Line 258: “However, it showed a significant increase (p-value < 0.05) to 39.24 × 107 conidia per cm2 after adding 6 µM biotin (Fig. 5a).” and Line 261: “However, it showed a significant increase (p-value < 0.05) to 1.03 × 107 conidia per cm2 after adding 6 µM biotin (Fig. 5c).”

Point 26:Line 280: Use a more objective term than "believed", such as "we suggest" or "we propose".

Response:Thank you so much for your comment. In the resubmitted manuscript, we have replaced “believed” with “suggested”. We are grateful for your professional suggestions to help us improve this manuscript.

Round 2

Reviewer 1 Report

The authors improved the introduction, methodology and results. Only one term (homogeneous recombination) mentioned in the methodology should be revised.

The authors described in more detail the introduction, methodology and results. Only one term (homogeneous recombination) mentioned in the methodology should be revised.

Author Response

We appreciate you very much for your positive and constructive comments on our manuscript. We have provided a detailed response to your conments. Please review the uploaded file. Meanwhile, in the re-submitted manuscript, revisions in the first round were shown in red font, and the revisions in the second round (the present round) were highlighted in yellow background.

Point 1:Please clarify the term "homogeneous recombination", perhaps the appropriate term is homologous recombination.

Response:Thank you very much for your comment. We greatly appreciate your professional advice. In Line 98 of the resubmitted manuscript, we have revised 'homogeneous recombination' to 'homologous recombination'.

Reviewer 2 Report

This work is highly relevant as it explores the importance of the rad6 gene in biotin metabolism and its impact on the development of Beauveria bassiana, with a particular focus on the conidiation process. Beauveria bassiana is a commercially used entomopathogenic fungus, where conidia represent the primary infective unit due to their greater resistance during field application compared to blastospores and mycelium. Therefore, a deep understanding of the conidiation process is crucial for improving the production of this entomopathogen.

Regarding the scientific content of the publication I have no comments.

I would only suggest reviewing the use of italics for gene symbols and scientific names (yeast, line 79), and ensuring consistency in the notation of liter abbreviations, the spacing between numbers and units, as well as the formatting of p < 0.05, although this falls under the editors' responsibility.

Author Response

We appreciate you very much for your positive and constructive comments on our manuscript. We have provided a detailed response to your conments. Please review the uploaded file. Meanwhile, in the re-submitted manuscript, revisions in the first round were shown in red font, and the revisions in the second round (the present round) were highlighted in yellow background.

Point 1:I would only suggest reviewing the use of italics for gene symbols and scientific names (yeast, line 79), and ensuring consistency in the notation of liter abbreviations, the spacing between numbers and units, as well as the formatting of p < 0.05, although this falls under the editors' responsibility.

Response:Thank you so much for your comment. We are grateful for your professional suggestions to help us improve this manuscript. In the resubmitted manuscript, we have removed the italicization of the word 'yeast' in Line 79. After checking the entire text, we have made modifications to the use of italics for gene symbols, the notation of liter abbreviation, the spaces between numbers and units in the manuscript, as well as the format of p<0.05, to ensure formatting consistency.

The modified contents are shown as follows (highlighted in yellow backgroud):

  • In Line 103: “..., the upstream (1541 bp) and downstream (1553 bp) fragments of the rad6..”;
  • In Line 111: “The putative disruption transformants were initially screened on CDA plates (200 μg/mL phosphinothricin or 15 μg/mL chlorimuron ethyl), ...”;
  • In Line 119: “..., at 25℃ on a 12:12 h light/dark cycle for 8 days, and then the concentration of conidia was adjusted to 107 conidia mL−1 for subsequent experiments.”;
  • In Line 125: “..., 100 μL aliquots of a 107 conidia mL−1 suspension were evenly spread onto the cellophane-covered SDAY and CDA plates, ...”;
  • In Line 130: “...then subjected to ultrasonic vibration in 1 mL of 0.02% Tween 80 liquid to quantify conidiation capability.”;
  • In Line 143: “DEGs were defined as downregulated (log2 R(Δrad6/WT) ≤ −1) or upregulated (log2 R(Δrad6/WT) ≥ 1) at a significance level of q (corrected p) < 0.05.”; 
  • In Line 150: “WT and the rad6 disruption strain were incubated under the same conditions described in 2.2, ... ”;
  • In Line 159: “..., 100 μL of conidial suspension at a concentration of 107 conidia mL−1 was uniformly inoculated into...”;
  • In Line 214: “All experiments were initiated by spreading 100 μLof a 107 conidia mL−1 ”;
  • In Line 215: “Asterisked bars in each group significantly differ from unmarked bars (Tukey’s HSD, p < 0.05).”;
  • In Line 240: “Differentially expressed genes (DEGs) are defined as log2 FC ≤ –1 or log2 FC ≥ 1 at the level of p < 0.05. The remaining genes are insignificantly affected (–1 ≤ log2 FC ≤ 1).”;
  • In Line 244:“p-value < 0.001 labelled ***, p-value < 0.01 labelled **, and p-value < 0.05 labelled *).”;
  • In Line 250:“p-value < 0.001 labelled ***, p-value < 0.01 labelled **, and p-value < 0.05 labelled *.”;
  • In Line 267: “Asterisked bars in each group significantly differ from unmarked bars (Tukey’s HSD, p < 0.05).”;
  • In Line 289: “Asterisked bars in each group significantly differ from unmarked bars (Tukey’s HSD, p < 0.05).”;
  • In Line 319: “Fluffy genes flbA–flbE are well-known activators of brlA in Aspergillus nidulans.”.
